# Effect of Environmental Concentration of Carbamazepine on the Behaviour and Gene Expression of Laboratory Rats

**DOI:** 10.3390/ani13132097

**Published:** 2023-06-24

**Authors:** Milena Santariová, Kateřina Zadinová, Hana Vostrá-Vydrová, Martina Frühauf Kolářová, Sebnem Kurhan, Helena Chaloupková

**Affiliations:** 1Department of Ethology and Companion Animal Science, Czech University of Life Science Prague, Kamýcká 129, 165 00 Prague, Czech Republic; vostrah@af.czu.cz (H.V.-V.); chaloupkovah@af.czu.cz (H.C.); 2Department of Animal Science, Czech University of Life Science Prague, Kamýcká 129, 165 00 Prague, Czech Republic; zadinova@af.czu.cz; 3Department of Veterinary Sciences, Czech University of Life Science Prague, Kamýcká 129, 165 00 Prague, Czech Republic; fruhauf_kolarova@af.czu.cz; 4Department of Food Science, Czech University of Life Science Prague, Kamýcká 129, 165 00 Prague, Czech Republic; kurhan@af.czu.cz

**Keywords:** carbamazepine, behaviour, laboratory rat, gene expression

## Abstract

**Simple Summary:**

Carbamazepine is an active compound in commonly used drugs for the treatment of epilepsy and other diseases. It is a very stable substance that remains in the environment, especially in waste water, but also in surface water. Because of this, it can pose a potential risk to both animals and humans. Although its effect in therapeutic doses has been described, the effects of low doses on the environment are unknown. We were interested in whether the consumption of food contaminated with carbamazepine can cause certain changes in behaviour and whether it simultaneously affects gene expression in higher vertebrates. The laboratory rats served as model organisms. The animals were divided into three groups, where one group orally received a high therapeutic dose of carbamazepine, the second received a low environmental dose, and the third control group received a solution without carbamazepine. The results of behavioural testing, which focused on cognitive abilities, anxiety-like behaviour, and social behaviour, did not reveal any changes; however, the expression of a certain gene on the brain of rats was affected by carbamazepine.

**Abstract:**

Carbamazepine (CBZ), an effective drug for epilepsy and other neurological diseases, and its metabolites are one of the most frequently detected substances in the aquatic environment. Although these are doses of very low concentrations, chronic exposure to them can affect the physiological processes of living organisms. This experiment may clarify if carbamazepine, under an environmental and a therapeutic concentration, can affect the behaviour of higher vertebrates, especially mammals, and gene expressions of *Ugt1a6* and *Ugt1a7* in the brain compared to the control group without exposure to CBZ. Three groups of thirteen rats were randomly formed, and each group was treated either with carbamazepine 12 mg/kg (therapeutic), carbamazepine 0.1 mg/kg (environmental), or by 10% DMSO solution (control). The memory, anxiety, and social behaviour of the rats were assessed by the test Elevated Plus Maze, the novel object recognition test, and the social chamber paradigm. After testing, they were euthanised and brain tissue samples were collected and analysed for mRNA expression of *Ugt1a6* and *Ugt1a7* genes. The tests did not show significant differences in the behaviour of the rats between the groups. However, there were significant changes at the gene expression level of *Ugt1a7*.

## 1. Introduction

Carbamazepine (CBZ) belongs to a group of drugs known as Antiseizure drugs (ASDs) [1,2]. This group of drugs is characterized by a reduction in excessive neuronal activity in the brain [2]. The therapeutic mechanism of action of CBZ is the blockade of presynaptic voltage-gated sodium channels and the control of excitatory synaptic transmission [3]. Because of its effects, CBZ is used as an effective antiseizure drug and also has beneficial effects in treating a wide range of non-epileptic disorders including schizophrenia and mood and anxiety disorders [4,5]. From a neuropsychological perspective, CBZ is known for its anticonvulsant, mood-stabilizing [6], and anxiolytic effects [7], for example, decreased aggressive behaviour in patients exhibiting epileptic or behavioural disorders [8]. 

Vast quantities of drugs, however, are excreted after human and veterinary medication as active substances of pharmaceuticals or their metabolites and subsequently enter the environment. Among these substances, carbamazepine is detected as one of the most common [9]. This is due to its high global production [10], but also to the fact that carbamazepine and its metabolites are resistant to microbial degradation and sorption onto sludge in present municipal wastewater treatment plants [11,12]. The distribution of pharmaceuticals in the environment copies the entire hydrological cycle; therefore, these substances are found not only in surface water, but also in subsurface and potable water [13]. There, they were detected in concentrations as high as 6.3 μg/L in wastewater, in average concentrations up to 2.3 μg/L in effluent samples, and in the range of 0.1–1 μg/L in groundwater [14] and in species tissues, including bivalves [15,16]. Some studies have already shown the potential uptake of CBZ by plants, which was irrigated with reclaimed wastewater and might further be used as feed for livestock [17,18]. Studies have previously demonstrated that CBZ is taken up by and accumulates in terrestrial plants with typical concentrations measured in a range from 2.9 to 67 ng/g in leaf material [19]. 

Although CBZ metabolites and degradation products are found in ecosystems at low levels far below the therapeutic doses used in medical practice, their long-term exposure can have a chronic effect on the organism [20]. In various aquatic organism species, it was indicated that environmental doses of CBZ, aside from physiological processes such as changes in reproductive system or hormonal levels [21], also affect behavioural changes such as lower stress reactions, higher swimming speed, and prolonged feeding behaviour [21,22]. Similarly, other pollutants are known to influence the behaviour of the several species (e.g., di-(2-ethylhexyl) phthalate aggravated the autism-like behaviour in mice [23]; polystyrene microspheres reduced the feeding behaviour in larvae of *Artemia salina* [24]). It was demonstrated that the exposure of larval-stage fish to an environmental dose of 1 μg/L of carbamazepine resulted in an increase in their spontaneous movements and a faster response to touch stimulation, and they overall became much more active than those not exposed to CBZ [25]. 

While the effect of environmental doses of carbamazepine on behaviour has been investigated in aquatic organisms, there is a lack of knowledge about the effect of the environmental presentation of CBZ on the behaviour of terrestrial vertebrates. However, the effect of a therapeutic dose of CBZ is well described in laboratory animals such as mice and rats [26,27,28]. A number of studies have demonstrated the effects of carbamazepine in reducing anxiety-like behaviour [7,29,30]; in the Elevated Plus Maze test, rats showed a greater percentage of entries into the open arms of an aperture and a higher percentage of time spent in these arms [7,31]. Regarding cognitive abilities in higher vertebrates, previous studies have reported that the administration of CBZ does not significantly impair performance on cognitive tasks [32,33] and poses a lower risk for learning and memory impairment than the administration of other ASDs [34]. Conversely, CBZ appears to enhance the performance of mice and rats in various learning and memory tasks [26,27]. It has been shown that the prolonged administration of CBZ led, in rats, to an improvement in nonspatial memory as tested through spontaneous object recognition [27] and in spatial memory tested in Morris’s water maze [28,35]. Other studies have demonstrated that CBZ improves memory in passive-avoidance tests, T-maze, and Y-maze [33]. In addition, CBZ is also thought to enhance habituation memory performance [33]. The question must therefore be asked whether even an environmental dose of CBZ can affect the behaviour of higher vertebrates. 

There is also a lack of information about the effect of carbamazepine on social behaviour, namely, how it can influence social interactions or the motivation to seek out social contact. In the case of aquatic organisms, [21] evaluated whether CBZ modifies shoaling behaviours of the fish *Jenynsia multidentata*, but no significant changes in shoaling behaviour due to CBZ exposure were demonstrated [21]. In human patients suffering from outbursts of anger or rage as a result of severe closed head injury, social behaviour improved when carbamazepine was given in doses of 400 to 800 mg daily for 8 weeks [36].

When an organism encounters a drug, a reaction also takes place at the level of gene expression. Therefore, inter-individual differences in the response to a drug in terms of a loss or reduction in its efficacy and toxicity are often related to a variability in the expressed levels of genes involved in the drug disposition process [37,38,39]. These differences may subsequently manifest themselves in both physiological and behavioural manifestations of the individual organism.

Nowadays, we know the genes involved in the pharmacokinetics and pharmacodynamics of drugs and their influence on the efficacy and safety of drugs. These genes also cause variabilities in the individual response to a particular drug [37,38]. Genes that also play a role in the brain include UDP-glucuronosyltransferases (UGTs). UGTs are phase II biotransformation enzymes that glucuronidate numerous endobiotic and xenobiotic substrates. The UGT1A subfamily is involved in the metabolism of endogenous compounds and various drugs [37]. UGT-mediated drug resistance can be associated with either congenital overexpression of the enzyme, referred to as intrinsic drug resistance, or induced expression of the enzyme, referred to as acquired drug resistance, which is observed when enzyme expression is induced by a drug or other factors such as food-derived compounds [40]. The antiepileptic CBZ has been described as having a strong effect on the functioning of some UGT isoforms [41]. It is the gene expression of the UGT subfamily that is responsible for the homeostasis and efficacy of each drug [42]. High expression of the *Ugt1a6* and *Ugt1a7* genes was observed in the rat brain. Therefore, CBZ can be expected to alter the expression in rat brain [41].

The aim of this study was to determine whether the low environmental concentration of carbamazepine, which is a known contaminant of individual components of the agroecosystem, affects the behavioural phenotype, especially the social behaviour, cognitive abilities, anxiety-like behaviour, and gene expression of *UGT* genes in the brain, while the laboratory rat was chosen as the model organism for this research.

## 2. Materials and Methods

The maintenance of all experimental animals was carried out in compliance with the current laws of the Czech Republic (Act No. 246/1992 coll. on Protection of animals against cruelty) and the European Union (EC Directive 86/609/EEC). The minimum number of experimental animals was used to achieve appropriate results. All procedures were approved by the Institutional Ethics Committee and were in accordance with NIH Animal Care Guidelines, nb. 63479/2016-MZE-17214.

### 2.1. Animals and Treatment

The study was provided in the Demonstrational and Experimental Workplace—Animal House in Animal Life Sciences in Prague. For the experiment, 39 adult males of Wistar rats from (Velaz s.r.o., Prague, Czech Republic) were used. The animals were housed in a group of 2–3 rats into a plastic box with a metal lid. Each box was equipped with a plastic house and lined with commercially supplied wood shavings litter that was replaced regularly. The area of the box was 1875 cm^2^. The rats were kept under standard laboratory conditions: temperature ±22 °C and humidity 30–60%. They were fed commercial pellets (An-Lab, Prague, Czech Republic) and allowed to drink tap water ad libitum. One light bulb fastened in the upper part of the room provided a constant illumination of about 40 lux at the level of the test apparatus.

The rats aged 24 weeks were divided according to the treatment, that is, according to the amount of CBZ that was administered to them. Carbamazepine treatments were prepared weekly at two different doses. The environmental dose was prepared with 2.52 mg of CBZ standard material, and the therapeutic dose was prepared with 302.4 mg of CBZ in 12.1 mL of DMSO and distributed into individual tubes for daily treatments. The CBZ solutions were stored at 4 °C and the tube content was defrosted before each use and completed to a volume of 18 mL with MiliQ water. The animals were consecutively given relevant CBZ concentrations at environmental and therapeutic levels at 0.1 mg/kg of body weight and 12 mg/kg of BW. Relevant CBZ concentrations were set based on their low and high accumulation rates in plants, which were irrigated with reclaimed wastewater and might further be used as feed for livestock [18,36].

The control group was treated with only 10% DMSO containing MilliQ water. CBZ was administrated using 4.8 mg CBZ/2 mL (therapeutic dose) and 0.04 mg CBZ/2 mL solutions, which were incorporated into the food mixture. The indicated concentrations corresponded to 12 mg/kg (4.8 mg/400 g of body weight) and 0.1 mg/kg (0.04 mg/400 g of body weight). The given CBZ dose was adjusted according to the rats’ body weights, which were recorded daily. The daily body weight gains of the rats is given in the Appendix A. The final DMSO concentration in the treatments was set as 10%.

The boxes with the rats were placed in three shelves one above the other. In order to influence the placement of the box at a specific height, all treatments were represented in each shelf.

The trial took place in the following phases:

1. Adaptation period (getting used to the environment, handling associated with the experiment, and eating the meat mixture) lasting 60 days. During the adaptation phase, the animals were handled daily. It is very well documented that gentle manipulation before an experiment serves to habituate them to the stresses associated with testing [35], and they were also trained once a day to remain individually in plastic boxes and to consume a meat mixture; later in the experimental period, it was a meat mixture with CBZ.

2. Experimental period for 14 days, during which rats were given an aqueous solution with CBZ (or without in the case of the control group) mixed in a meat mixture, between 9 and 10 AM. During carbamazepine administration, rats were placed individually in small plastic boxes, where they were given food containing carbamazepine. They were checked during this time, and after consuming the entire dose, the rats were returned to the cage where they were housed.

3. Behavioural testing during the next five days, when the rats aged 26 weeks were subjected to consecutive behavioural tests—for details, see the schedule (Figure 1). During the entire time, the rats were fed a mixture of carbamazepine solution mixed with meat. Behavioural testing was performed between 10:00 a.m. and 6:00 p.m. 

The behaviour of the tested animal during all behavioural tests was recorded by one camera located above the arena or maze at a height of 240 cm above the ground, and the recorded behaviour was analysed by a special SW for analysing animal behaviour, the Observer XT (Noldus, The Netherlands) system.

### 2.2. Behavioural Tests

#### 2.2.1. Elevated plus Maze

The construction has a “+” cross shape and consists of five zones: two open illuminated arms (50 × 10 cm), two closed shaded arms (50 × 10 × 40 cm), and a central open platform (10 × 10 cm). It serves to assess the effects of pharmacological agents on anxiety and to observe fear-related behaviour. The entire EPM apparatus was placed 103 cm high above the floor. 

The rat was placed in the centre of the cross and then allowed to move freely through the maze. The movement was monitored by a camera placed above the maze. The recording took 5 min. The number of entries into open arms, the number of entries into the closed arms, and the total time spent in open arms and the total time spent in closed arms were monitored. The centre of gravity of the animal was chosen as the point on the body determining the visit to the zone. 

These behavioural measurements were consequently used to calculate indices that were entered into the statistical analysis as the dependent variables; the ratio OA [the ratio of time spent in the open arms to the number of entries into those arms OAT/OAE] indicates how long the rat spent in the open arm during one entry, and the percentages of open arm entries (OAEs) and open arm time (OAT) were expressed by the ratio OAT/(OAT+CAT) [OAT = the time spent in open arms and the ratio OAE/(OAE+CAE), OAE = the number of entries to the open arms, CAE = the number of entries to the closed arms]. Rat anxiety had a direct relation with the ratio (a lower ratio indicated more anxious rats) [7]. 

After each trial, all arms and the centre area were cleaned with 70% ethanol as an efficient odour removal agent to prevent bias based on olfactory cues [43]. We were able to thus conduct the tests under controlled conditions regarding olfactory cues.

#### 2.2.2. Novel Object Recognition Task

This is a test in which the tested animal recognizes a new object. The level of spontaneous exploratory behaviour of the experimental animal that naturally recognizes new objects in familiar surroundings is evaluated. The animal pays increased attention to the new object, and this interest spontaneously initiates active exploratory behaviour. This test is used to evaluate working memory and attention, and possibly also the preference for new stimuli. In the case of memory damage, the animal stops focusing primarily on the unknown object and looks at both the known and the unknown object for approximately the same amount of time. The training chamber was an open field apparatus (45 × 40 × 60 cm^3^) made of plastic. 

On the first day, all animals were submitted to a habituation session during which they were placed in the empty open field and left to freely explore the arena for 5 min. The test comprised two five-minute trials. In the first trial “training session”, two round metal objects (heavy enough that they cannot be manipulated) were placed in each adjacent corner of the box. The rat was placed in the centre of the test arena at the same distance to both objects. After five minutes, the animal was removed. During the second trial “choice session”, an object was replaced with a new object and the animal was returned to the arena. Between trials, the objects were washed with a 70% ethanol solution. The animals were tested after 10 min of retention. The original object presented during the second task was a duplicate of the sample in order to avoid olfactory trails. From rat to rat, the role (sample or new object) and the position of the two objects during the second task were counterbalanced and randomly permuted. These precautions were taken to reduce object and place preference effects. The time for which the animal devoted itself to the new object and the old object (sniffing, panning, and staying in the immediate vicinity of the object) was measured.

Exploratory behaviour was considered and counted every time the animal was directed to the object, touching it with its paw or nose, or sniffing it. We subsequently derived behavioural parameters calculated through several indices (Table 1). The habituation index (NOR ih) was calculated as the difference between the time spent exploring both objects in the training session and the time spent exploring both objects in the choice session. A higher index value shows that the rats will spend a shorter time exploring both objects in the choice session than in the training session, and it is an indicator of habituating memory. 

The discrimination index expresses the discrimination between new and familiar objects. It uses the difference in exploration time for the initial object and new object (NOR D1) and can also be calculated as the difference in exploration time for the familiar object, but then dividing this value by the total amount of exploration of the new and familiar objects (NOR DI). This result indicates more time spent with the new object or more time spent with the familiar object [44,45,46]. 

The recognition index (NOR RI) is the time spent investigating the new object relative to total object investigation, and it is the main index of retention [47,48].

In order to verify the initial or new object preference, we also used a visit index (NOR VI), which was calculated as a ratio of the number of new object contacts to the total number of contacts of both objects in the choice session.

#### 2.2.3. Social Preference Test

The Social Preference Test allows us to evaluate the sociability and social novelty preference of the observed individual by monitoring its preference for either an empty chamber or a stimulating animal. The test arena was a plastic box consisting of three adjacent chambers (60 cm × 40 cm × 20 cm) separated by two clear plastic dividers and connected by an open doorway (4 cm × 3 cm). During the pre-training phase, the test subject was placed in the middle chamber of the apparatus. The sliding doors were opened to allow the rat to familiarize itself with the entire arena (the duration of time in each of the two outside stimulus compartments was noted with stopwatches). In this phase, the cages in the side chambers remained empty. Once the animal had explored the entire area, it was placed back into the central chamber with the doors to the side chambers closed. 

##### First Phase: “Sociability”

After the pre-training phase, the test animal was placed in the central trigger chamber and allowed to acclimatize in the given space. The stimulus-unfamiliar rat was then placed in one of the lateral cages in a randomly selected side. The stimulus animal was also familiarized with the cage at least 24 h before the actual test. This was important so that the stimulus animal remained calm during the test and was not engaged in aggressive interactions.

For each tested individual, the chamber in which the stimulus animal was placed was randomly changed. The stimulation cage in the second chamber then remained empty. The door to the side chamber was opened, and the subject was allowed to examine the apparatus.

During the testing phase, the behaviour of the tested individual was monitored. Data were collected on the amount of time spent in each of the chambers, the number of chamber entries, the time spent sniffing at each cage, and the number of contacts with a stimulus animal. Stimulus Contact was recorded when the subject rat made contact with any part of the wire cup or the stimulus rat.

The test phase was finished after 10 min.

##### Second Phase: “Preference of Social Novelty”

Immediately after the first test, the test subject was placed back into the central trigger chamber, and the doors to the side chambers were closed. The second stimulation animal was placed in the second cage (empty until then). At this moment, both side chambers contained one original stimulus animal and a second new unknown stimulus animal. After the placement of the second stimulus animal, the partition doors were opened and the second test phase was performed for 10 min as described for the first test phase. 

In order to eliminate the influence of the side on the preferences of the subjects, the side of the presented stimulus was systematically counterbalanced between subjects.

### 2.3. Euthanasia of Animals and Sampling

The rats were euthanized at the end of the study with simultaneous sampling for subsequent analyses. The rats were euthanized individually and in a separate room to minimize signs of stress in other rats. Rats were put under general anaesthesia in an inhalation mask with isoflurane (Isoflutek 1000 mg/g, LABORATORIOS KARIZOO, S.A., Spain). The euthanasia itself was performed by an intracardiac injection of 0.5 mL of T61 (Intervet International B.V., Unterschleissheim, Germany) with the active ingredients embutramidum, mebezonii iodidum and tetracaini hydrochloridum. After euthanasia, decapitation was performed at the atlantooccipital junction and cerebellar tissue (100 mg) was collected via the foramen magnum using a curettage spoon in 19 male rats. All samples were immediately immersed in RNAlater TM Stabilization Solution (Thermo Fisher Scientific, Waltham, MA, USA) and stored at −20 °C until processing.

### 2.4. Primer Design, Isolation of RNA, and qPCR

Specific transcript variant PCR primers for reference genes (*Actb*; *Gadph*; *Ppia*) and target genes (*Ugt1a6*; *Ugt1a7*) were taken from previously published sources (see Table 2). The selection of reference genes was performed by using three commonly used reference genes in rats according to Chen et al. [49] and Feria-Romero et al. [50].

The samples of the brain (cerebellum) were removed from RNA later and rinsed several times in PCR ultra-water. An amount of 100 mg of cerebellum sample was homogenized in 1 mL of Purezol (Qiagen, Hilden, Germany), and the total RNA was isolated using the Aurum Total RNA Fatty and Fibrous Tissue Kit (Bio-Rad Laboratories Inc., Hercules, CA, USA). For more detail, see [51].

qPCR was performed on the CFX96 Touch Real-Time PCR Detection System (Bio-Rad Laboratories Inc.) in 10 μL reaction volumes using 5 μL of KAPA SYBR FAST qPCR master mix (KAPA Biosystems, Wilmington, MA, USA), 1 μL of primer mix (5 μM each), and 2 μL of cDNA (10-fold diluted). The qPCR profile was 95 °C for 3 min, followed by 32 cycles of 20 s at 95 °C and 30 s at the optimal annealing/elongation temperature (Table 2) with detection of fluorescence. A melting curve was generated by heating the samples from 70 to 95 °C in steps of 0.5 °C for 5 s with fluorescence detection. All reactions were performed at least two times. The quantification cycle values of the target gene were converted to raw data based on the efficiency of the qPCR. PCR efficiencies per gene were between 97.7% and 102.5%. The Cq values (calibrated averages of the technical duplicates) were transformed into quantities, by using the delta-Cq formula with the highest expression level set to 1. The relative expression was formulated as a ratio of the transformed Cq-value from the target gene to the geometric mean of 3 reference genes (*Actb*, *Gadph*, and *Ppia*). The whole RT-qPCR experiment was performed according to the MIQE guidelines [52]. 

**Table 2 animals-13-02097-t002:** Primers sequences and qPCR assay details for target and reference genes.

Gene	Primers	Primer Sequences (5′–3′)	Amplicon (bp)	T_a_ ^1^ (°C)	R^2^ ^2^	PCR Efficiency (%)	Reference
** *Actb* **	F	ATCGCTGACAGGATGCAGAAG	108	62	0.996	102.5	[49]
	R	AGAGCCACCAATCCACACAGA					
** *Gadph* **	F	AGGGCTGCCTTCTCTTGTGAC	101	58	0.997	99.4	[49]
	R	TGGGTAGAATCATACTGGAACATGTAG					
** *Ppia* **	F	AAGGTGAAAGAAGGCATGAG	76	56	0.997	101.2	[50]
	R	CCGCAAGTCAAAGAAATTAGAG					
** *Ugt1a6* **	F	GGGAGAATCCAAATACTACAGGAG	100	60	0.999	101.8	[53]
	R	CAGCAAAGTGGTTGTTCCCAAAGG					
** *Ugt1a7* **	F	CAGACCCCGGTGACTATGACA	73	61	0.997	97.7	[53]
	R	CAACGTGAAGTCTGTGCGTAACA					

^1^ Annealing temperature; ^2^ Coefficient of determination.

#### Statistical Analyses

All data were analysed with the aid of SAS software (SAS Institute Inc., Cary, NC, USA). The goodness of fit of each model (homoscedasticity, normality of errors, independence) was checked by visually inspecting residuals using plots. Results with a *p*-value of less than 0.05 (*p* ≤ 0.05) were considered statistically significant. We applied the Linear Mixed Model (LMM-REML) to test for differences in the fixed factor “treatment”. Data were analysed by a Mixed Linear Model Procedure (PROC MIXED) with repeated measurements by the following model:$$y_{ij}={treatment}_{i}+a_{j}+e_{ij}$$
where *y_ij_* is a dependent variable (dependent variables are described in the individual behavioural test). Treatment is a fixed effect of the *i* (*i* = therapeutic, environmental, and control), a is a random effect of the *j* batch, and e is a residual error. A post hoc test (Tukey) was used to compare the adjusted means. We used the covariance structure that fit best according to the Tukey–Kramer information criterion.

The original data of the gene expression study were log-transformed prior to statistical analysis. These data are normally distributed for all nutrition groups. The normality of variables was checked using the Kolmogorov–Smirnov test. The data were analysed with one-way analysis of variance (ANOVA). The results are presented as the least squares means (LSMs) and root-mean-square error (RMSE). Differences between LSMs were determined by Duncan’s test (*p* < 0.05). The statistical model was
*y_ij_* = *μ* + *g_i_* + *e_ij_*
where *y_ij_* is value of the traits, *μ* is the overall mean, *g_i_* is the effect of the diet group (*i* = C, E, T), and *e_ij_* is the random residual error.

## 3. Results

### 3.1. Elevated plus Maze

The treatment had no significant effect on the OA (F_2,34_ = 0.47, *p* = 0.63), OAE (F_2,34_ = 0.38, *p* = 0.69), and OAT (F_2,34_ = 0.70, *p* = 0.51). For details, see Table 3.

### 3.2. Novel Object Recognition Test

The treatment had no significant effect on the rats’ behaviour on NOR_ih (F_2,35_ = 0.64, *p* = 0.53), NOR_D1 (F_2,35_ = 0.42, *p* = 0.66), NOR_DI (F_2,35_ = 0.12, *p* = 0.88), NOR_RI (F_2,35_ = 0.12, *p* = 0.88), and NOR_VI (F_2,35_ = 0.68, *p* = 0.51). Differences between individual treatments are presented in Table 4.

### 3.3. Social Chamber Test

The treatment had no significant effect on all variables measured during the Social Chamber Test (see Table 5 for Phase 1 and Table 6 for Phase 2).

### 3.4. The Effect of CBZ on Gene Expression

In the case of the mRNA expression of both genes (*Ugt1a6* and *Ugt1a7*), differences were found between the groups. For both genes, the lowest mRNA expression was observed in the therapeutic group. The highest *Ugt1a6* mRNA expression was observed in the control group (Figure 2); however, the difference was not significant (F_2_ = 1.85, *p* = 0.189). A significant difference (F_2_ = 3.74, *p* = 0.046) was observed for *Ugt1a7* mRNA expression between the environmental and therapeutic groups (Figure 3); however, the control group did not differ from the environmental and therapeutic groups. In both cases (*Ugt1a6*, *Ugt1a7*), the expression was almost twice as high in the control and environmental groups compared to the therapeutic group.

## 4. Discussion

Carbamazepine is a very persistent pharmaceutical product and one of the most frequently detected in the environment. While its therapeutic effect and side-effects on the behaviour of high vertebrates has been investigated, the effect of environmental doses of carbamazepine and its potential risk has yet to be clarified. Our aim was to evaluate the effects of the low environmental concentration of carbamazepine on the behaviour of laboratory rats. We focused on the potential changes of three groups of behaviour: cognition–memory, emotion–anxiety, and social-behaviour–interest in social interaction. 

The animals were consecutively given relevant CBZ concentrations at environmental and therapeutic levels at 0.1 mg/kg of body weight and 12 mg/kg of BW, and the control group was treated with only 10% DMSO containing MilliQ water. After fourteen days, the animals were tested in three types of standardized behavioural tests.

Neither the environmental nor the therapeutic dose of CBZ affected the behaviour of rats in all behavioural tests compared to the control group.

### 4.1. Elevated plus Maze Anxiety Test

Carbamazepine (CBZ) is an anticonvulsant and mood-stabilizing drug that has also been shown to have anxiolytic effects [54]. The Elevated Plus Maze is a standard behavioural test used to assess the effects of various anxiolytic drugs. This test maze assumes that rats prefer dark, enclosed spaces, because they have an innate fear of heights and open space. An increased willingness to enter and remain in the open arm reflects anti-anxiety behaviour [55], so anxiolytic drugs are thought to increase the time spent in the open arms of the test maze. Given that CBZ has been shown to have anxiolytic effects, we hypothesized that rats given CBZ would spend more time in the open arms than control rats, or that they will show a higher number of entries into these arms. 

Based on the OAT, OAE, and OA indices, we evaluated the percentage of time spent in the open arms, the number of entries into the open arms, and the average time spent in the open arms after entry. A higher index value means that the rats spent more time in the open arm and entered the open arm more, which is considered to be a lower manifestation of anxious behaviour.

In our study, the percentage of entries into open arms did not statistically differ between treatment groups, but this parameter may not show the actual anxiolytic effect of the CBZ as stated in the study [7]. The anxiolytic effect is probably much better demonstrated by the time the rats spent in the open arms, because the percentage of time spent in the open arms is more sensitive to drug effects than the number entries in the Elevated Plus Maze [56]. However, the percentage of time each group spent in the open arms did not differ significantly in our study either. At the level of behavioural manifestations, specifically anxiety, the administration of carbamazepine in therapeutic doses or in environmental doses has not been proven. 

Therefore, our results are not in complete agreement with other studies [7,56]. Rats given an intraperitoneal injection of carbamazepine CBZ (5–40 mg/kg) increased the percentage of entries into the open arms and the percentage of time spent in them [31]; also in the study of [7], a systemic administration of dose of 40 mg/kg increased the percentage of time that the rats spent in the open arms, which suggests the anxiolytic properties of CBZ. However, our study differed in the method of CBZ application: oral application in food in the present study as compared to intraperitoneal injections [7]. 

Another possible factor that influenced the behaviour of the rats, and therefore the results of the study, was that the rats were regularly handled every day. This manipulation was carried out in order to minimize stress in the planned tests. The rats were thus able to handle the transfer without problems to the plastic boxes, where they were individually fed with a meat mixture with a carbamazepine solution [57,58]. 

### 4.2. Novel Object Recognition Test

A number of studies have addressed whether carbamazepine poses a risk of cognitive impairment of higher vertebrates [28,59,60]. The effect of therapeutic doses of carbamazepine was mostly studied in epileptic animals, in which the side-effects of this drug were evaluated [27,34,61]. For that reason, we decided to determine whether the environmental concentration of CBZ can affect cognitive abilities, focusing on certain types of memory. Because the treatment of carbamazepine has been shown to affect habituating and discrimination memory [27], we used the NORT as the optimal test to assess these abilities in rats administered therapeutic and environmental doses of carbamazepine.

NORT aims to investigate the memory of animals and their ability to recognize a new object from a familiar one. The principle is based on the animal’s natural tendency to pay more attention to a new unfamiliar object, thus requiring the presentation of the familiar object to exist in the animal’s memory [62]. As useful indicators of the rat exploring the object, we used the behaviour during which the animal sniffed or touched the objects with its nose or paws. Through the calculation of the index of habituation, discrimination [63], recognition [47], and the visiting index [64], we evaluated cognitive abilities manifested by habituating memory, discriminative memory, and the recognition of objects. The positive mean value of the indices reflected the animal showing more interest in an object with which it has already had the opportunity to become familiar.

CBZ administration, in both environmental and therapeutic concentration, was shown not to impair the cognitive abilities we tested. Carbamazepine-treated animals did not show compromised habituation in the NORT, allowing the conclusion that carbamazepine does not affect procedural memory. CBZ has also not been shown to impair discrimination memory. 

Given some findings that carbamazepine administration can improve cognitive abilities, specifically habituating memory and discriminative memory in rats [27], we hypothesized that the group of rats receiving the CBZ (therapeutic dose 12 mg/kg) could show better performance. For the group receiving a dose as low as 0.1 mg, we did not expect any effect on their cognitive performance. The results of our study did not show that 0.1 mg/kg of CBZ increased exploratory behaviour in rats, but treatment with 12 mg/kg of CBZ did not produce any change in the exploration of objects. The exploration time of both objects did not differ statistically between the groups. This finding is not fully consistent with the results of a preliminary study [27], where it was proven that the prolonged treatment with CBZ in epileptic rats induced a significant increase in object discrimination during the choice session, with this being interpreted as CBZ having a positive effect on a simple object-discrimination task. However, in the latter study, the dose of CBZ administered was much higher than in our study, specifically 40 mg/kg, while our therapeutic dose was 12 mg/kg. It is also necessary to realize that this study involved epileptic animals and these results may be associated with the psychotropic effects of CBZ. Nevertheless, our results are partially consistent with the results of other studies in that CBZ does not significantly impair memory and learning [32,65].

### 4.3. Social Preference Test

Most rodents, including the rats, demonstrate strong social communities with relationships that are achieved by the physical contact between individuals [66]. For this reason, rats provide an optimal model for measuring sociability. Our aim was to evaluate whether an exposure to CBZ can modify sociability such as the pursuit of social contact and preferences of social novelty. Sociability was defined as the tendency to approach and remain proximal to an unfamiliar conspecific vs. an avoidance of the stranger rat by remaining in the empty chamber or exploring. Within the three-chamber test, most laboratory rats show preference for the contact with a rat over spending time in an empty chamber, and in a latter phase, a preference for a new rat placed in the previously empty cage [67,68,69]. Given that CBZ is a drug with anticonvulsant and mood-stabilizing effects [6,65], we expected that rats would not display aggressive behaviour toward each other. In the three-chamber test, we assumed the rats would show an enhanced investigation/preference for social stimuli and, in the second phase of the experiment, a preference for novel social stimuli. 

Contrary to our prediction, it does not seem that the administration of carbamazepine, either in environmental or therapeutic concentrations, changes social behaviour and social preferences in any way. Neither during sociability nor during the preference of social novelty did we detect differences between groups of rats treated with the therapeutic and environmental dose of CBZ compared to the control animals. But, there is a considerable lack of studies on the issue of the influence of CBZ on social behaviour, and therefore it is not possible to compare our results with the results of other experiments. The possible explanation of these results can be the age of the animals. The rats were tested at an age of six months, which is later compared to other studies, where the same testing paradigm was used [63,70,71]. At different ages, animals can react differently, even though they have a physiological reaction according to measurable parameters. Aging can influence the behaviour and physiology of emotional expression to stressful stimuli [72]. The younger animals can show a significantly higher frequency of avoidance as well as greater exploratory activity, while the older subjects are less behaviourally responsive and show a shift toward control by the sympathetic nervous system, as indicated by a lower heart rate variability [73]. The question is also what effect intensive handling, which was carried out throughout the experiment, could have had on the rats’ behaviour. In addition, they were trained to be able to eat the CBZ mixture individually in special plastic boxes and this was again an occasion when the rats were handled and experienced repeated mild stress [74]. However, it can be concluded that CBZ does not significantly affect the sociability of rats.

### 4.4. Effect of CBZ on Gene Expression in Rat Brain Tissue

The expression of drug metabolizing enzymes and drug transporters in a number of organs and tissues determines local and systemic drug exposure and the resultant pharmacological and toxicological effects. Therefore, interindividual variation in the drug response in terms of a loss of drug efficacy as well as drug toxicities is often related to a variability in the expressed levels of genes involved in the drug disposition process. Among the many mechanisms involved in the regulation of gene expression, it appears that the constitutive, induced, and repressed expression of drug disposition genes is largely under transcriptional control [53]. In the case of our study, the therapeutic group differed from previously described experiments primarily in that it was intended to represent the dose commonly administered to patients as a drug. This group had the lowest expression of both genes; however, a statistically significant difference was detected only for the *Ugt1a7* gene. While these results are inconsistent with a previous study [53], this study included rats significantly younger (8 weeks x 21 weeks); furthermore, the rats were fed a significantly higher dose for a shorter period of time (100 mg/kg; 7 days). For this reason, we assume that the body of the animal in the study of Asai et al. [53] shows an immediate response to encountering the drug and tries to minimize its effects. In the present study, the situation was different; the animals were exposed to the drug for a longer period, so we assumed that the dose of drug would overcome the body’s response and begin to affect it, which is the primary goal of drug administration [40]. On the other hand, the environmental group, which for the same time period ingested only a minimum of CBZ, i.e., as much as is normally found in nature, showed an expression comparable to the control group.

## 5. Conclusions

Oral administration of carbamazepine did not show an effect on the social behaviour, cognitive abilities, and anxiety-like behaviour of the laboratory adult rats at either therapeutic or environmental concentration compared to the control group without CBZ. The result of mRNA expression can be interpreted in that the level of CBZ commonly available in nature does not induce a stronger reaction in the organism even with long-term use compared to the control group, because the organism is able to cope with it due to the action of UDP-glucuronosyltransferases. Conversely, a dose of the drug administered as medication for a prolonged period will overcome the physiological defences of the organism and will have an effect on the organism. This was confirmed by the decrease in mRNA expression for both genes studied.

## Figures and Tables

**Figure 1 animals-13-02097-f001:**
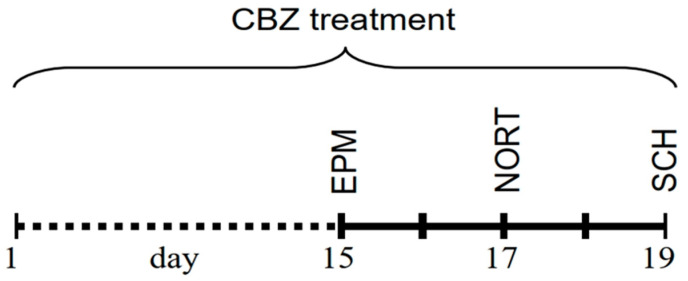
Time axis illustrating the experimental design of this study. Each point on the axis represents one day; points with numbers represent testing days; dotted points represent days with no experiments. EPM—Elevated Plus Maze test, NORT—novel recognition test, SCH—social chamber test.

**Figure 2 animals-13-02097-f002:**
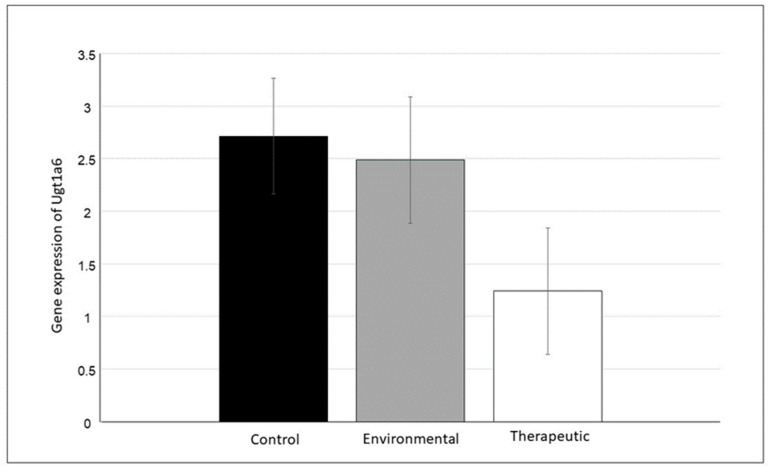
Effect of CBZ treatment on the changes in the expression of *Ugt1a6* mRNA in the rat brain. Data are presented as LS means ± S.E. Non-significant differences between treatments, *p* = 0.19.

**Figure 3 animals-13-02097-f003:**
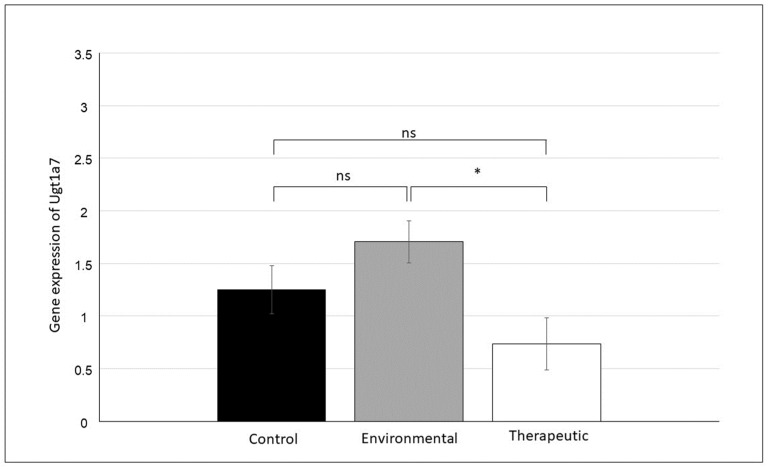
Effect of CBZ treatment on the changes in the expression of *Ugt1a7* mRNA in the rat brain. Data are presented as LS means ± S.E., * *p* ≤ 0.05, ns = non-significant difference.

**Table 1 animals-13-02097-t001:** Definition of the behavioural measurements in the novel object recognition test.

Variables	Definition
e1	time spent in the zone of both objects in (phase 0) training session (5 min)
e2	time spent in the zone of both objects (new old) in (phase I) choice session (5 min)
Teo	exploration time of the initial object in (phase I) choice session (5 min)
Ten	exploration time of the new object in (phase I) choice session (5 min)
VISo	number of visits to the starting object in phase I choice session (5 min)
VISn	number of visits to the new object in phase I choice session (5 min)
NOR_ih	habituation index of exploratory behaviour (phase 0) training session vs. (I) choice session NOR_ih = e1 − e2
NOR_D1	discrimination index between new and familiar object NOR_D1 = TZn − Tzo
NOR_DI	discrimination index between new and familiar object NOR_DI = D1/e2
NOR_RI	recognition index NOR_RI = TZn/e2
NOR_VI	visit index (new vs. familiar object) NOR_VI = VISn/VISn + VISo

**Table 3 animals-13-02097-t003:** Effect of the treatment on the behaviour of rats in the Elevated Plus Maze. Data are presented as LS means ± S.E.

Item	Therapeutic	Environmental	Control	*p*-Value
OAE ^1^	0.317 ± 0.053	0.363 ± 0.054	0.325 ± 0.053	Non-sign.
OAT ^2^	0.241 ± 0.068	0.344 ± 0.070	0.293 ± 0.068	Non-sign.
OA ^3^	0.035 ± 0.007	0.044 ± 0.007	0.042 ± 0.007	Non-sign.

^1^ OAE—the percentages of open arm entries; ^2^ OAT—the ratio of time spent in the open arms to time spent in the closed arms; ^3^ OA—the ratio of time spent in the open arms to the number of entries into those arms.

**Table 4 animals-13-02097-t004:** The effect of the treatment on the behaviour of rats in the novel object recognition tests. Data are presented as LS means ± S.E.

Item	Therapeutic	Environmental	Control	*p*-Value
NOR_ih ^1^	0.077 ± 0.022	0.061 ± 0.022	0.089 ± 0.022	Non-sign.
NOR_D1 ^2^	−0.015 ± 0.017	−0.002 ± 0.017	−0.004 ± 0.017	Non-sign.
NOR_DI ^3^	−0.062 ± 0.274	−0.057 ± 0.274	0.014 ± 0.274	Non-sign.
NOR_RI ^4^	0.469 ± 0.137	0.471 ± 0.137	0.507 ± 0.137	Non-sign.
NOR_VI ^5^	0.439 ± 0.086	0.432 ± 0.086	0.502 ± 0.086	Non-sign.

^1^ NOR_ih—habituation index of exploratory behaviour; ^2^ NOR_D1—discrimination index between new and familiar object; ^3^ NOR_DI—discrimination index between new and familiar object; ^4^ NOR_RI—recognition index; ^5^ NOR_VI—visit index (new vs. familiar object).

**Table 5 animals-13-02097-t005:** Effect of the treatment on the behaviour of rats in the Social Chamber Test during Phase 1.

Variable	Measurement	Therapeutic	Environmental	Control	F-Value (df)
Contact with UNF ^1^ rat_1	Total number	11.22 ± 1.24	11.01 ± 1.24	12.14 ± 1.24	0.52 (2,35)
	Duration, s	0.44 ± 0.04	0.44 ± 0.04	0.40 ± 0.04	0.41 (2,35)
Entry to the 1st Section	Total number	3.85 ± 0.34	4.31 ± 0.34	4.08 ± 0.34	0.47 (2,35)
	Duration, s	0.71± 0.05	0.70 ± 0.05	0.72 ± 0.05	0.07 (2,35)
Entry to the middle section	Total number	4.00 ± 0.35	4.54 ± 0.35	4.46 ± 0.35	0.68 (2,35)
	Duration, s	0.78 ± 0.07	0.89 ± 0.07	0.88 ± 0.07	0.76 (2,35)

^1^ UNF—unfamiliar rat.

**Table 6 animals-13-02097-t006:** Effect of the treatment on the behaviour of rats in the Social Chamber Test during Phase 2.

Variable	Measurement	Therapeutic	Environmental	Control	F-Value (df)
Repeated contact	Total number	5.84 ± 0.75	5.85 ± 0.75	4.08 ± 0.75	1.88 (2,35)
	Duration, s	0.23 ±0.06	0.22 ± 0.06	0.21 ± 0.06	0.05 (2,35)
Repeated entry to the 1st Section	Total number	2.22 ± 0.38	2.92 ± 0.40	2.37 ± 0.38	1.24 (2,34)
	Duration, s	0.45 ± 0.06	0.44 ± 0.06	0.34 ± 0.06	0.94 (2,34)
Repeated entry to the middle section	Total number	4.60 ± 0.67	4.71 ± 0.67	5.06 ± 0.67	0.16 (2,35)
	Duration, s	0.13 ± 0.03	0.20 ± 0.03	0.15 ± 0.03	1.79 (2,35)
Contact with UNF ^1^ rat_2	Total number	6.67 ± 0.92	7.20 ± 1.00	8.08 ± 0.88	0.63 (2,31)
	Duration, s	0.17 ± 0.04	0.16 ± 0.04	0.24 ± 0.04	1.80 (2,31)
Entry to the 2nd Section	Total number	2.33 ± 0.35	2.60 ± 0.38	2.54 ± 0.33	0.15 (2,31)
	Duration, s	0.44 ± 0.05	0.40 ± 0.06	0.49 ± 0.05	0.77 (2,31)

^1^ UNF—unfamiliar rat.

## Data Availability

The data are not publicly available, due to privacy and ethical reasons.

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
