# Peer review of "Effect of Environmental Concentration of Carbamazepine on the Behaviour and Gene Expression of Laboratory Rats"

_animals, 2023, doi:10.3390/ani13132097_

Round 1
Reviewer 1 Report
Santariová et al. investigate the important possibility that long term exposure to bioaccumulation-equivalent levels of carbamazepine might affect the behavior and levels of drug metabolizing enzymes Ugt1a6 and Ugt1a7 in the brain of vertebrates. Using rats, they conclude that there were no effects on behavioral indices, but there was an impact of therapeutic-equivalent exposure on gene expression of Ugt1a7.
The experiments seem technically sound. I have some small concerns about the expression of their results and conclusions.
Have the authors considered checking general activity in these rats, similar to the reported elevations in aquatic animals? Although not pure activity indices, they might check total arm entries and total distance traveled in the elevated plus maze and maybe general activity in the other tests.
How do the control animals perform on the NOR and Social Chamber Tests? I am a bit concerned that the NOR is not sensitive enough to capture recognition memory since the NOR_RI of the Controls seems to be neutral at 0.507 – it seems like they explore both objects equally. The authors should discuss this.
Does “Duration, s” in Tables 5ab mean average durations or total?
In the Social Chamber Test, the authors compare absolute measures between treatment groups, but a critical measure in Phase 2 is the comparison of interaction with the unfamiliar rat versus the familiar rat, is it not? How do the Control rats perform on this measure? Do they perform as predicted, showing preference for interacting with the unfamiliar rat?
In Figure 3 it is not quite clear what only a and only b mean.
Toward the end of the Discussion and the Conclusions, the authors should be more precise about what they mean by “the body… tries to minimize its effects” and “overcome the body’s response and begin to affect it” (and how is that the primary goal of drug administration?). Also clarify “overcome physiological defences of the organism and will have an effect on the organism” and explain how this has anything to do with Ugt1a7.
Tolerance due to neural adaptations is a common way for drug responses to change after repeated exposure. Could the seeming lack of behavioral effects be due to tolerance? What is known about the tolerance profile of CBZ?
Author Response
Dear reviewer,
Thank all of you for valuable comments.
On the basis of the review, we explain below the changes made in the paper. In this letter, we refer to individual paragraphs in the reviewer’s comments.
Changes made in manuscript are visible in manuscript with tracked changes. We supplemented some information in the introduction, in methods. Also, some literature has been added in introduction.
We are very grateful for the careful reading of the manuscript and very valuable comments, which enabled us to improve the manuscript substantially and for the interest to read our article by four reviewers.
Yours sincerely,
Milena Santariová

Reviewer 2 Report
Comments are listed here for the author’s consideration to further improve the quality and overall impact of the manuscript.
1. Authors mention “high therapeutic dose of carbamazepine (12 mg/kg)” and “low environmental dose (0.1 mg/kg)”. How did the authors determine these carbamazepine (CBZ) doses as high or low? therapeutic or enviromental doses?? Explain the basis for choosing these doses and include references.
2. Why did each experimental group consist of 13 rats?
3. Line 46: Anticonvulsants is a wrong term. Change it by Antiseizure drugs (ASDs), throughout the entire manuscript. Please refer to the following reference:
French JA, Perucca E. Time to Start Calling Things by Their Own Names? The Case for Antiseizure Medicines. Epilepsy Curr. 2020 Mar;20(2):69-72. doi: 10.1177/1535759720905516.
4. Line 47. “This group of drugs is characterized by a reduction of excessive nerve activity in the brain”. Reduction of excessive nerve activity??? Or neuronal activity?? Correct the term.
5. Line 50. “CBZ is used as an effective antiepiepticum…” What it means effective antiepiepticum??? Correct the term.
6. In the introduction, indicate the relevance of the UGT genes (Ugt1a6 and Ugt1a7), what they are, their physiological function, etc., specifically in the brain of rats.
7. Authors mention that CBZ was administered orally by given food containing carbamazepine. But how did the authors ensure that the desired levels of CBZ were actually reached in the rat in a period of 1 hour (9-10 AM)? or that the rat actually consumed what was necessary to say that the indicated dose was administered?
8. Do not the behavioral tests performed interfere with each other since they were performed almost immediately after on the same animal? Some tests required a 24-hour habituation period prior to testing.
9. In addition, animals continued to be treated with CBZ during behavioral tests. So, by day 15, there were 15 administrations of CBZ on rats. On day 17, animals had already 17 CBZ´s administrations. Finally on day 19, there werea 19 total administrations of CBZ. That is, each behavioral test evaluated had a different number of administrations of CBZ that could influence the results. Authors should discuss this possibility.
10. The authors should also discuss the effects of the doses of CBZ used in their study, why they chose these doses (therapeutic and environmental), why that route of administration, why that period of administration (14 days), in which it was dissolved CBZ, and if all these factors could affect the results obtained.
11. Line 303. “Subsequently, 0.5 to 1 ml of whole blood was collected from the v. cava cranialis.” Why was this sample collected? The processing of the sample or results are not indicated. Include it.
12.Line 306. “Liver, kidney and brain tissue samples were then collected for further analysis.” The processing of the samples (Liver and kidney) or results (Liver and kidney) are not indicated. Include it.
13. Line 318. “The 100 mg of cerebellum sample was homogenized…”, Was brain or cerebellar tissue analyzed? Correct it.
14. Line 331. Mention why these 3 reference genes (Actb, Gadph and Ppia) were used.
15. The authors should discuss what is the physiological implication that the expression levels of the evaluated genes are modified in the brain of the rat, treated with CBZ.
16. Line 458. “The mechanism of its action is the potentiation of GABA receptors.” Authors must specify the main mechanisms of pharmacological action of CBZ.
17. Line 484. “…a systematic administration of dose 40 mg/kg…”, a systematic administration??? Or systemic administration? Correct it.
18. Authors mention at conclusions that long-term exposure of low levels of CBZ doesn´t have a stronger reaction on the organism, but how did authors determine that 14 days of daily CBZ administration is comparable to a period of long-term exposure? Explain it and and include references.
19. Also, how did authors determine that the level of CBZ (environmental dose 0,1 mg/kg) is the level commonly available in nature? Explain it and and include references.
Authors must correct their conclusions by being specific with the results they obtained.
Requires revision of English.
Author Response

(The authors gave the same response as above.)

Reviewer 3 Report
My comment are:
- in the introduction section please include the following reference, as many pollutants can influence the behavior of several species.
DOI10.3390/app11083352
- 10.1016/j.intimp.2020.107323
- line 122: "The minimum number of 122 experimental animals was used to achieve appropriate results." How was it calculated?
-On what basis were the doses of CBZ calculated?
- The daily weight data should be displayed in the manuscript.
-line 308-312 please details better
-Please clarify how the gene expression data was calculated, as the authors did not use the delta-delta Ct method.
To provide clearer results, expressing the data as fold change over the control would be more informative.
Author Response

(The authors gave the same response as above.)

Round 2
Reviewer 2 Report
Change antiepileptic drug by antiseizure drug, in the introduction section.
Minor editing of English language required.
Author Response
Dear reviewer,
thank you very much for your recommended correction. We have changed antiepileptic drug by antiseizure drug, in the introduction section.
Kind regards
Milena Santariová